

# A study of collider signatures for two Higgs doublet models with a Pseudoscalar mediator to Dark Matter

Jonathan M. Butterworth[1], Martin Habedank[2]$^\star$, Priscilla Pani[3] and Andrius Vaitkus[1]

**1** Department of Physics & Astronomy, UCL, Gower St., WC1E 6BT, London, UK
**2** Department of Physics, Humboldt University, Berlin, Germany
**3** DESY, Hamburg and Zeuthen, Germany

$\star$ martin.habedank@physik.hu-berlin.de

## Abstract

Two Higgs doublet models with an additional pseudoscalar particle coupling to the Standard Model and to a new stable, neutral particle, provide an attractive and fairly minimal route to solving the problem of Dark Matter. They have been the subject of several searches at the LHC. We study the impact of existing LHC measurements on such models, first in the benchmark regions addressed by searches and then after relaxing some of their assumptions and broadening the parameter ranges considered. In each case we study how the new parameters change the potentially visible signatures at the LHC, and identify which of these signatures should already have had a significant impact on existing measurements. This allows us to set some first constraints on a number of so far unstudied scenarios.

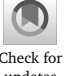

# 1  Introduction

A Dark Matter (DM) model involving two Higgs doublets and an additional pseudoscalar mediator [1,2] (2HDM+a) has been the subject of several searches at the LHC [3–5]. It provides the simplest theoretically consistent extension of DM simplified models with pseudoscalar mediators [6]. In contrast to models with scalar mediators, which are heavily constrained by direct detection measurements, pseudoscalar mediators offer the advantage of being currently safe from those constraints due to the spin-dependent nature of their direct detection cross section, making them inviting candidates to address the subject of DM at colliders. [7]

The considered model contains a number of more-or-less free parameters, essentially masses and mixing angles of the bosons and a coupling to the DM candidate. This leads to a particularly rich phenomenology, dominated by the production of the lightest pseudoscalar or the heavier Higgs boson partners via loop-induced gluon fusion, associated production with heavy-flavour quarks or associated production with a Standard Model (SM) Higgs or gauge boson. Depending on the values of the parameters, very diverse characteristic signatures can be produced, including the traditional DM signature of missing transverse momentum ($E_T^{\mathrm{miss}}$), but also a range of SM-only final states. The relative importance of these final states varies strongly as the parameters of the model change. This leads to numerous potentially observable final states at colliders, many of which remain largely unexplored. The purpose of this paper is to explore some of them using existing LHC measurements.

For a given scenario, one expects a well-designed search to give optimal sensitivity to a targeted signature. However, because of the rich particle content of the model there may be contributions from other signatures, even in the benchmark scenarios typically considered in searches. Some of these signatures may contribute to 'SM-like' cross sections which have already been measured[1]. In this study we use CONTUR [8] to examine the sensitivity of ATLAS, CMS and LHCb particle-level (i.e. unfolded) measurements available in Rivet 3.1.1 [9] to a two Higgs-doublet model with a pseudoscalar mediator and DM particle.

The paper is structured as follows. First, we discuss the Herwig calculations used, the advantages and limitations of the CONTUR method, and the default parameters of the model. Then we revisit the benchmark scenarios proposed by the LPCC DM Working Group [2] and report the sensitivity of the measurements to these[2]. We then extend the explored parameter space away from the benchmark region; first, we relax the assumption that the exotic Higgs bosons are all degenerate in mass. Next, we vary the mass of the DM particle and the mediator, with a view to exploring regions where the correct DM relic density can be obtained from this model alone. After that, we vary both the mixing angle between the two neutral CP-odd weak eigenstates ($\sin\theta$) and the ratio of the vacuum expectation values of the Higgs doublets ($\tan\beta$). Finally, we relax the strict imposition of the alignment limit.

In each case we study how the new parameters change the potentially visible signatures at the LHC, and identify which of these signatures should already have had a significant impact on existing measurements. This allows us to set some first constraints on a number of so far

---

[1]And indeed may also contribute to control regions used in searches.
[2]This work follows on from a preliminary study performed as part of the Les Houches 2019 workshop [10].

unstudied scenarios, and to suggest some future directions for study. These are summarised in the final section.

## 2   Methodology

We use CONTUR 1.2[3] to scan over various parameter planes of the model, generating all $2 \to 2$ processes in which a Beyond the Standard Model (BSM) particle is either an outgoing leg, or an $s$-channel resonance, with Herwig [12]. This procedure omits some potentially significant $2 \to 3$ processes, such as $pp \to t\bar{t}X$ and $pp \to thX$, where $X$ is a BSM particle. These processes were studied using dedicated runs with Madgraph5_aMC@NLO [13], and found not to contribute significant additional sensitivity[4]. The calculations are leading order and Herwig includes also leading order loop processes that dominate the production for the DM mediator and the exotic Higgs bosons. It factorises the production of the particles from their decay using the narrow width approximation, so does not include interference terms with SM processes.

No matching or merging between the BSM matrix elements and the parton shower is being used in Herwig[5]. Therefore, because the parton shower can add jets above some minimum transverse momentum $k_\perp^{\min}$, there is an element of double counting between, for example, $gg \to H \to XY$ diagrams (where one or both of $X, Y$ are BSM states) with an additional hard gluon radiation from the parton shower, and $gg \to Hg$ diagrams where the $H$ decays to $XY$. The default value of $k_\perp^{\min}$ in Herwig is 20 GeV. A scan of this value over 10 GeV to 160 GeV indicated a relatively rapid fall in sensitivity between 10 GeV and 20 GeV, and a slower fall above this, presumably driven initially by the reduction in double-counted events and then by the loss of valid events as $k_\perp^{\min}$ increases to higher values which the parton shower will not populate. Since for this BSM model we work at leading order, we set $k_\perp^{\min} = 50$ GeV to be on the conservative side.

All the cross sections quoted have been calculated with Herwig 7 [15, 16] and with Madgraph5_aMC@NLO [13] and were found in good agreement.

We note that interference effects between SM $t\bar{t}$ production [17–21] have a significant effect on the shape of the mass distribution of the $A \to t\bar{t}$ decay, for example, and as already mentioned these effects are not taken into account by Herwig. However, none of the measurements used are specifically hunting for resonances; in general they are inclusive $W$+jet or top measurements, and so are unlikely to be very sensitive to differences in shape.

CONTUR identifies parameter points for which an observably significant number of events would have entered the fiducial phase space of the measurements, and evaluates the discrepancy this would have caused under the assumption that the measured values, which have all been shown to be consistent with the SM, are identical to it. This is used to derive an exclusion for each parameter point, taking into account correlations between experimental uncertainties where available. The speed of the CONTUR method, the inclusive approach of Herwig in generating all processes leading to BSM particle production, and the variety of measurements available, allow us to broaden the range of parameters considered into some interesting regions, away from the usual benchmarks. Nevertheless, the starting point is the default parameters settings derived from [2], and detailed in Table 1. In each section that follows, the parameters are as given in this Table unless explicitly stated otherwise.

---

[3]We are considering a search for supersymmetric quarks in the 0-lepton channel [11] in addition to the LHC measurements used by default.

[4]The studies are summarised in Appendix A.

[5]Although the issue is well studied for higher-order QCD SM processes, see for example [14].

Table 1: Default parameter settings used in the 2HDM+a model. The masses of all the exotic Higgs bosons are indicated as $M_A$, $M_H$, $M_{H^\pm}$, the mass of the DM candidate is $M_{DM}$, the coupling of the pseudoscalar mediator $a$ to DM is $g_{\chi_d}$, $\theta$ is the mixing between the two neutral CP-odd weak eigenstates, and $\sin(\beta - \alpha)$ is the sine of the difference of the mixing angles in the scalar potential containing only the Higgs doublets. Finally, $\lambda_i$ are the quartic couplings of the Higgs potential.

| $M_H, M_A, M_{H^\pm}$ | $M_a$ | $M_{DM}$ | $\tan\beta$ | $\sin\theta$ | $g_{\chi_d}$ | $\sin(\beta - \alpha)$ | $\lambda_3, \lambda_{P1}, \lambda_{P2}$ |
|---|---|---|---|---|---|---|---|
| 600 GeV | 250 GeV | 10 GeV | 1 | 0.35 | 1 | 1 | 3 |

## 3 Benchmarks with degenerate exotic scalar masses

We first focus on two parameter scans, those of Fig. 19 in Reference [4]: the $(M_a, M_A)$ scan and the $(M_a, \tan\beta)$ scan. In these scans, which follow the recommendations of the LPCC DMWG [2], the masses of all the exotic Higgs bosons ($A$, $H$, $H^\pm$) are degenerate, the mass of the DM candidate $M_{DM} = 10$ GeV, the coupling of the pseudoscalar mediator $a$ to DM is unity, $\sin\theta = 0.35$ where $\theta$ is the mixing between the two neutral CP-odd weak eigenstates, and we set $\sin(\beta - \alpha)$, the sine of the difference of the mixing angles in the scalar potential containing only the Higgs doublets, to unity, meaning we are in the aligned limit so that the lightest mass eigenstate has SM Higgs couplings, and the quartic couplings are all set to $\lambda_i = 3$ (see Table 1). The results are shown in Fig. 1.

In the $(M_a, M_A)$ scan (Fig. 1a,c) the overall sensitivity is the combined result of relatively marginal (1-2 $\sigma$) contributions to a wide range of cross-section measurements. One of the few measurements involving missing energy, which was in fact targetted at $ZZ \rightarrow \nu\bar{\nu}\ell^+\ell^-$ [22] and uses only 7 TeV data, is sensitive at low $M_A$ and $M_a < M_H - M_Z$, as can be seen in Fig. 1c. This region overlaps, but is smaller than, the dilepton + $E_T^{miss}$ searches discussed in [4], which uses more luminosity and higher-energy collision data. The sensitivity is principally due to the $H \rightarrow aZ$ decay, which has a branching fraction of $\approx 30\%$ in this region.

Over much of the rest of the 95% excluded region, ATLAS jet substructure measurements [23] are most sensitive, especially in the hadronic $W$ selection; however, several other measurements, notably those aimed at $W$ or $Z$ plus jets, make comparable contributions. These final states generally arise from the production of all the exotic Higgs bosons, with subsequent decays either to top quarks or directly to $W$ bosons[6]. For example for $M_A = M_H = M_{H^\pm} = 435$ GeV and $M_a = 250$ GeV, the production cross-section for the CP-odd Higgs boson $A$ is about 6 pb and its dominant decay channels are in $t\bar{t}$ (85%) and DM pairs (15%). In addition, the production cross section for the CP-even Higgs boson $H$ is 3 pb, and it decays into a top-pair with a branching ratio of 88%, while the second dominant decay mode is $H \rightarrow aZ$ (10%).

In the $(M_a, \tan\beta)$ scan (Fig. 1b,d), most of the plane is excluded for $\tan\beta < 1$. This is again largely due to analyses involving $W$ boson production, especially now the CMS measurement of semi-leptonic $t\bar{t}$ decays, which give rise to lepton-plus-$E_T^{miss}$-plus-jet final states [24]. As before, these events come from decays of the exotic Higgs bosons. A similar scan was conducted in Ref. [25], reporting compatible sensitivities from recasting searches, most notably from a CMS search for heavy Higgs bosons decaying to a top quark pair [18].

The $(M_a, \tan\beta)$ scan presented in Fig. 1 has been extended to consider a wider range for the $\tan\beta$ parameter with respect to Ref. [4]. It is worth mentioning that for $\tan\beta > 50$, the width of the CP-even Higgs boson predicted by this model exceeds 20% of its mass for $M_a < 350$ GeV. As discussed in the previous section, the use of the narrow width approximation means the calculation of the BSM signal will be increasingly unreliable in these regions.

---

[6]As discussed in Ref. [10], we only consider measurements without data-driven control regions and where any $b$-jet veto is implemented not only at detector level but in the fiducial cross section definition.

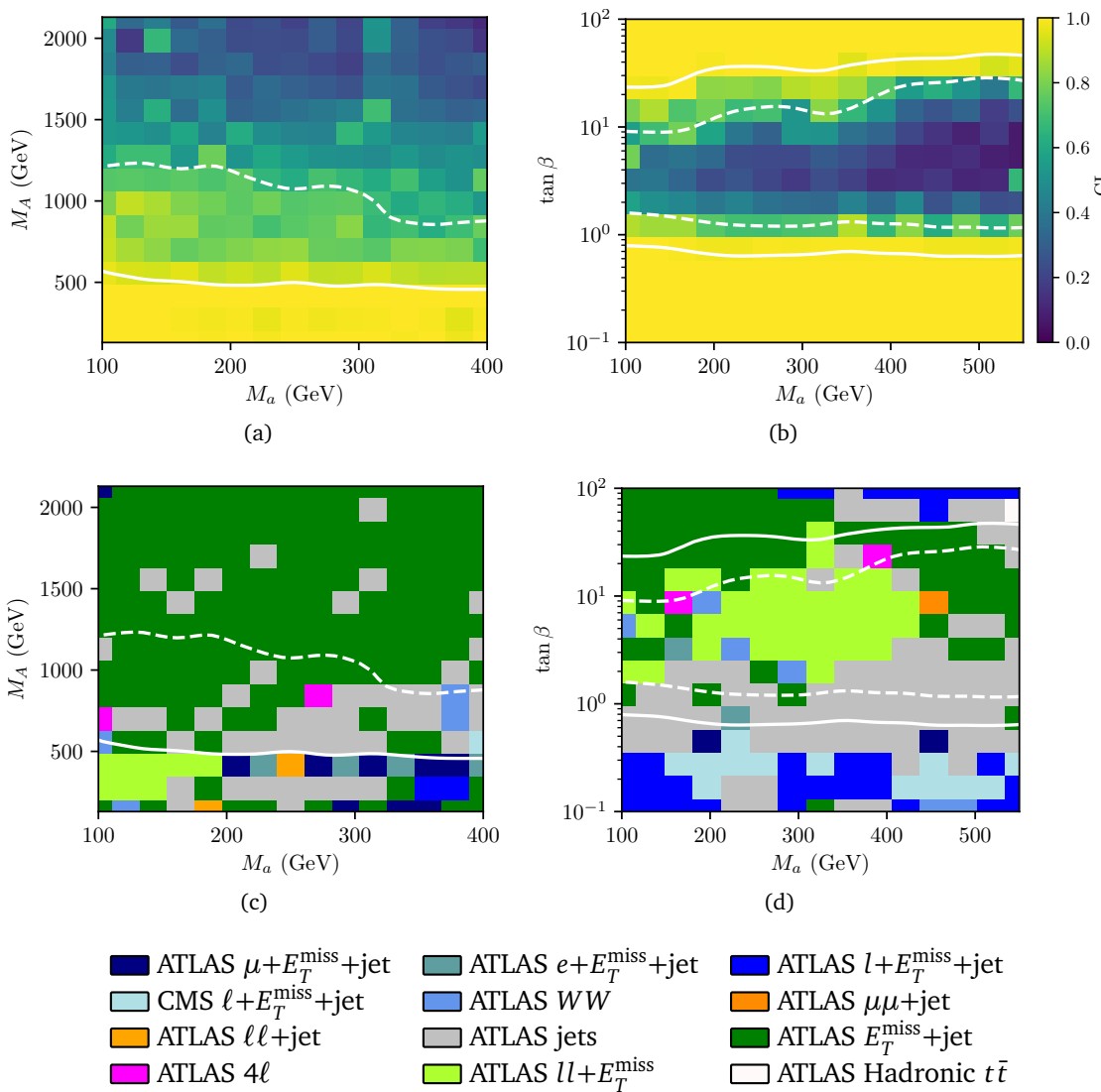

Figure 1: CONTUR scans over the parameter planes from [4] in the (left) $(M_a, M_A)$ and (right) $(M_a, \tan\beta)$ planes. The top row shows the sensitivity map in terms of CLs, with 95% (68%) excluded contours indicated by solid (dashed) white lines. The bottom row shows the analysis which contributes the most to the sensitivity of each signal point.

Overall, the SM measurements are highly complementary to searches in constraining the parameter space, and extend the coverage for $\tan\beta > 10(40)$ for $M_a = 100(500)$ GeV. The sensitivity in this region of the parameter space is driven mainly by $E_T^{\mathrm{miss}}$ +jet [11, 26] final states. This sensitivity comes from $bb$ and $gb$ initiated production, which is enhanced for higher $\tan\beta$ values. There are also contributions from processes such as $tH^\pm \to t\bar{t}b$ [27] which contribute to top final states, for example boosted top pairs [28].

## 4  Non-degenerate masses

The imposition of $M_H = M_A = M_{H^\pm}$ is a somewhat arbitrary choice; while one might expect none of the Higgs masses to be far from the electroweak symmetry-breaking scale, some variation may occur and this can have a significant impact on collider signatures. Relaxing this hypothesis was previously studied in the context of exotic Higgs searches in $t\bar{t}Z$ and $tWb$ final states [29]. There are other constraints on the masses however, as discussed in Ref. [1, 29]. Flavour and perturbativity constraints apply at low values of $\tan\beta$, with a constraint that $\tan\beta \geq 0.8$ for $M_{H^\pm} = 750$ GeV from $B$-meson observables, and weaker constraints from LHC searches. In the following, $\tan\beta = 1$ will be imposed (unless it is varied in a scan), so these constraints are always satisfied. Electroweak precision measurements also imply that $H$ and $A$ can only differ significantly in mass if either $M_H = M_{H^\pm}$ or $M_A = M_{H^\pm}$; both of these cases will be studied separately. In the case of $M_A = M_{H^\pm}$, there are additional constraints on the value of $\sin\theta$ from the custodial-symmetry breaking, generally favouring lower values. Finally, as discussed in [29], when the $A$ and $H$ masses differ by several 100 GeV, the width of the heavier of the two will get large due to decays to $H^\pm W^\mp$.

In this work, we will study the impact of the degeneracy assumptions using three benchmarks:

- $(M_a, \tan\beta)$ scan assuming $M_{H^\pm} = M_H = 750$ GeV, $M_A = 300$ GeV. This reproduces the parameters of "Scenario 4" of Ref. [1] and further assumes $\sin\theta = 1/\sqrt{2}$, $M_{\mathrm{DM}} = 1$ GeV.

- $(M_H, M_A)$ scan assuming $M_{H^\pm} = M_H$ and varying $M_a$ masses of (100,200,300,500) GeV.

- $(M_H, M_A)$ scan assuming $M_{H^\pm} = M_A$ and varying $M_a$ masses of (100,200,300,500) GeV.

All three benchmarks follow the LPCC DMWG convention for the other parameters (see Table 1).

### 4.1  Case 1

In [1], studies were performed for parameter points in which the degeneracy of the exotic Higgs boson masses was broken. These studies interpreted a number of LHC searches, including mono-jet, mono-Higgs and invisible Higgs searches. Their "Scenario 4" corresponds closely to our first benchmark above, the only difference being the fact that the quartic couplings were set to zero, which has negligible impact on the results. The comparable CONTUR scan is shown in Figure 2, along with the original figure.

Again, the absence of many $E_T^{\mathrm{miss}}$ measurements, especially mono-Higgs, reduces the sensitivity, visible in this case at low $M_a$ and moderate $\tan\beta$. However, $E_T^{\mathrm{miss}}$+jet, top, and $W$ measurements do disfavour the region $\tan\beta \lesssim 1$. There is also some sensitivity at high $\tan\beta$ (where our scan is extended to higher values) coming from $Z$, diphoton and $E_T^{\mathrm{miss}}$ measurements.

### 4.2  Case 2

Continuing with the charged Higgs mass $M_{H^\pm}$ equal to the heavy CP-even Higgs mass $M_H$, we now scan over the range 200 GeV to 2200 GeV, and scan $M_A$ independently over the same range. This two-dimensional scan was repeated for a few values of $M_a$ in the range 100 GeV to 500 GeV, and the results are shown in Fig. 3.

At low $M_a$ (Fig. 3a) the model is disfavoured in the $M_{H^\pm} = M_H < 500$ GeV region for all values of $M_A$, principally due to the $\ell\ell + E_T^{\mathrm{miss}}$ measurement [22]. This comes from exotic Higgs decays to $aZ$, with the $a$ then decaying to DM. As $M_a$ is increased, this sensitivity is reduced, as these decays are suppressed, and $H \to b\bar{b}$, a signature with larger SM background, dominates.

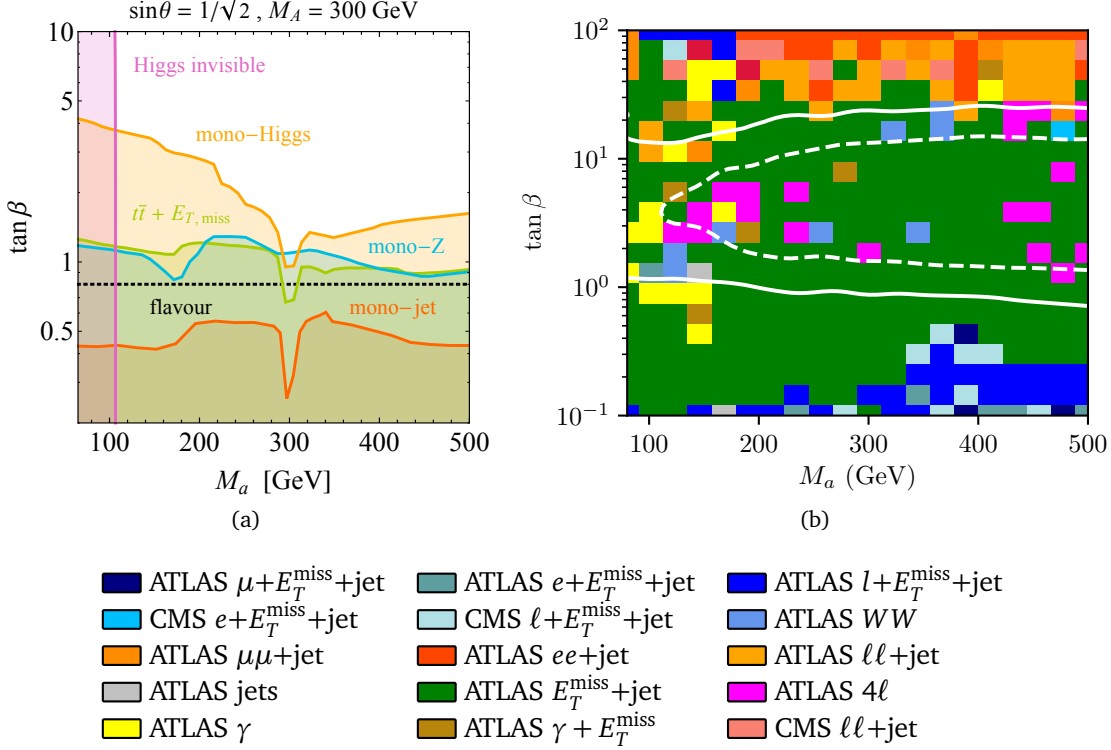

Figure 2: (a) Sensitivity scan from [1] (see text), (b) CONTUR scan over the same parameters, but extended to higher $\tan\beta$. The 95% (68%) excluded contours are indicated by solid (dashed) white lines; the colours indicate the most sensitive set of measurements.

At low $M_A$ there is reasonable sensitivity for all the $M_a$ values considered, coming from a variety of signatures. At low $M_{H^{\pm}} = M_H$, dilepton+(b−)jet measurements [30,31] contribute; in this region the $A$ decays to $HZ$ about a third of the time, with the $H$ decaying mostly to $b\bar{b}$. There are also contributions from missing energy and a lepton – essentially, top and $W$+jet cross sections; the charged Higgs bosons decay dominantly to $t\bar{b}$ at low $M_A$ for all the $M_a$ values considered. Finally, for $M_a$ not too high, low $M_A$ and high $M_H$, diphoton analyses have strong sensitivity; in this region the $A$ decays dominantly to $ah$, with the $h \to \gamma\gamma$ [32] (and also four-lepton final states from $h \to 4\ell$ [33]) becoming visible.

Over the rest of the plane, where the overall sensitivity is low, ATLAS jet substructure measurements [23] and $E_T^{\text{miss}}$+jet measurement [26] seem to offer the best chance of an eventual observation.

## 4.3 Case 3

The results of the scan in which the charged Higgs is degenerate with the heavy pseudoscalar $A$ are shown in Fig. 4. In this case the constraints from custodial symmetry-breaking [1] come into play and exclude the majority of the plane already. The general features of the CONTUR exclusion are similar to those in Section 4.2, in that the sensitivity is greatest when at least one of the new bosons has a mass of around 500 GeV or below. There are differences in the detailed reasons for this, however.

The dilepton+$E_T^{\text{miss}}$ signature still plays a role at low $M_a$ and $M_H$, although at high $M_{H^{\pm}} = M_A$ the $W$ or $Z$+jet signatures become more sensitive. The Higgs-to-diphoton measurements also

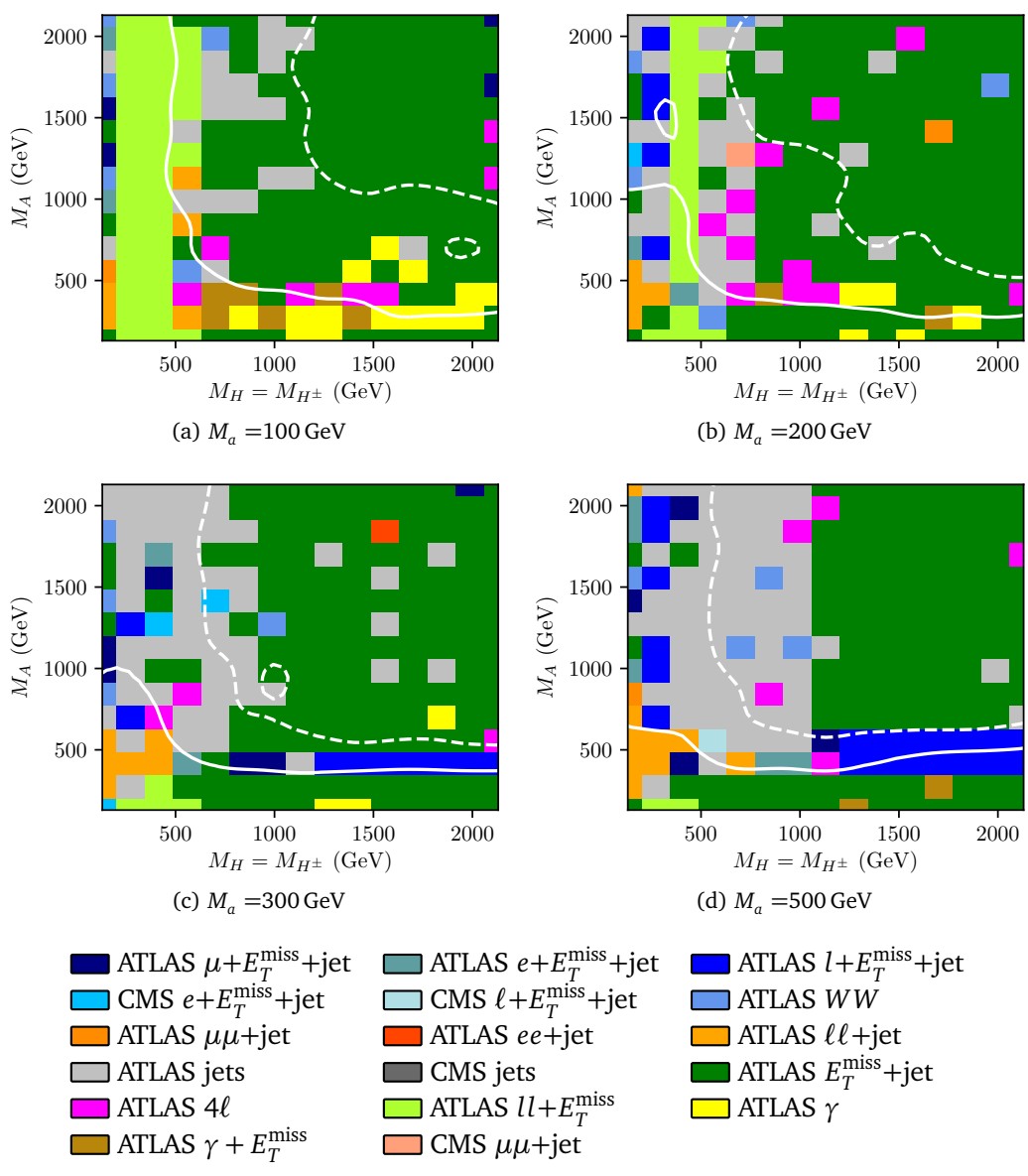

**Figure 3:** Sensitivity scan in $M_A$ vs $M_H = M_{H^\pm}$ for different $M_a$. The 95% (68%) excluded contours are indicated by solid (dashed) white lines; the colours indicate the most sensitive set of measurements. All other parameters are as given in Table 1.

play a role again, not only at low $M_A$ as before, but also at low $M_H$, since $A \to ah$ makes a significant contribution.

As in Section 4.2, over the rest of the plane, where the overall sensitivity is low, ATLAS jet substructure measurements [23] and $E_T^{\mathrm{miss}}$+jet measurements [26] receive the largest contributions to their cross sections.

For all the $M_a$ values considered, our constraints exclude practically all the previously allowed regions away from $M_A = M_H$, for the chosen value of $\sin\theta = 0.35$.

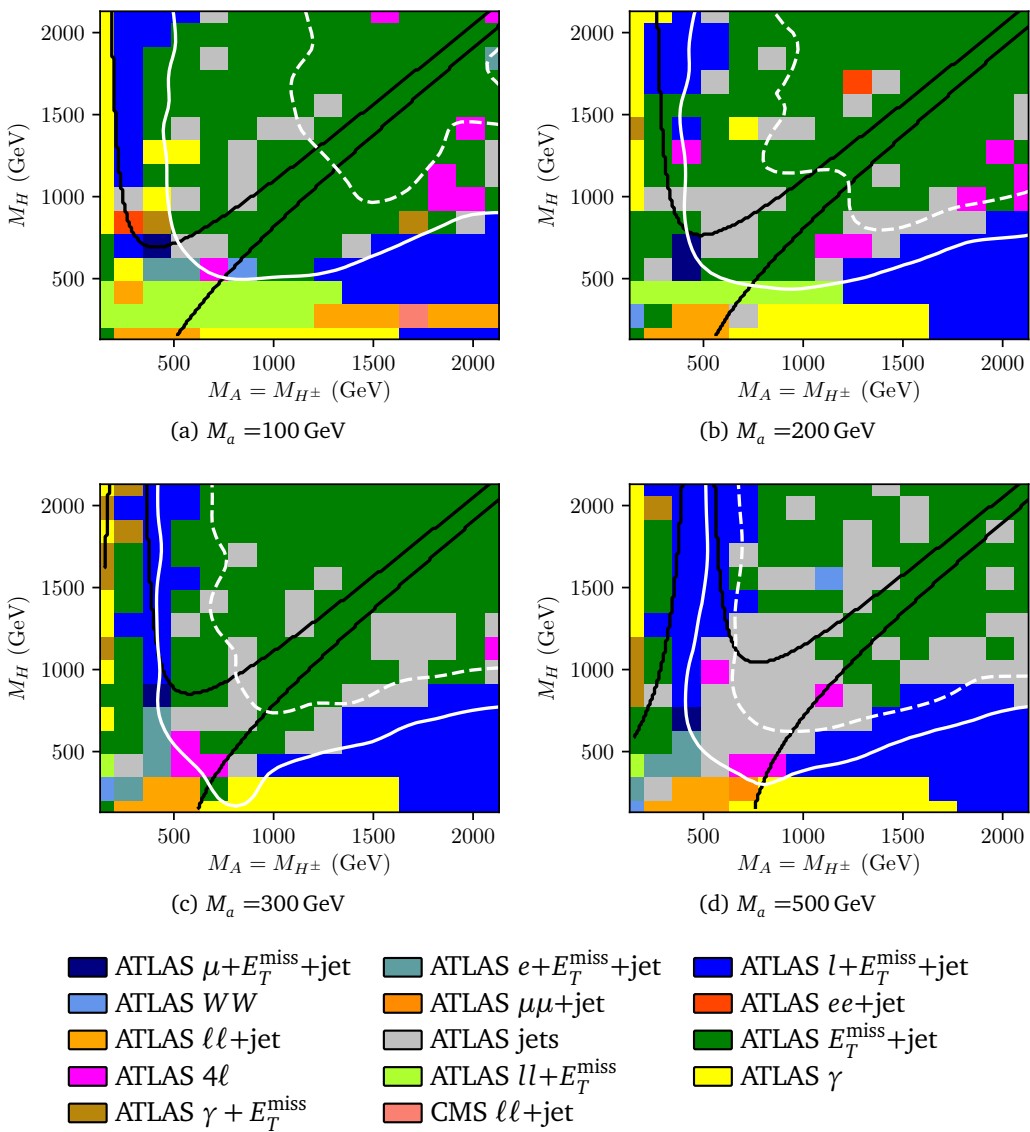

Figure 4: Sensitivity scan in $M_H$ vs $M_A = M_{H^\pm}$ for different $M_a$. The 95% (68%) excluded contours are indicated by solid (dashed) white lines; the colours indicate the most sensitive set of measurements. All other parameters are as given in Table 1. The region outside the black contours is already excluded by electroweak precision constraints on custodial symmetry.

## 5 Varying the DM mass

In cosmological models where the relic density of DM is determined at thermal freeze-out, and in the absence of further additional scattering mechanisms and particles, the parameters of the model determine this density. Astrophysical observations favour a value of $\Omega h^2 = 0.12$ [34, 35]. The most relevant parameters in calculating $\Omega h^2$ from the model are the masses of the DM, $M_{\rm DM}$, and of the mediator, $M_a$, although other parameters and the relations between them do have an influence. As discussed in [2], for the benchmark choice of $M_{\rm DM} = 10\,{\rm GeV}$ this constraint disfavours the benchmark. However, for other values of $M_{\rm DM}$ the correct relic

density can be obtained without significantly influencing the collider phenomenology.

In Figure 5a, we show the benchmark scan in $M_{\text{DM}}$ vs. $M_a$, where in [2] it was shown that for most values of $M_a$, the correct relic density is obtained for $M_{\text{DM}}$ values of around 200 GeV. While very little of the plane is currently excluded at 95% confidence level, there is sensitivity at the 68% level in vector-boson-plus-jet measurements (including the hadronic decays studied in the ATLAS jet substructure measurement), indicating that future precision measurements will have a significant impact.

Figure 5b shows a scan over $\tan\beta$ for $M_{\text{DM}}$ values between 1 GeV and 1 TeV. In this plane (for the chosen values of the other parameters, particularly $M_a = 250$ GeV), $M_{\text{DM}}$ needs to be between 100 GeV to 150 GeV to achieve $\Omega h^2 = 0.12$. We see that the sensitivity of the measurements shows little dependence on $M_{\text{DM}}$, and values in the range $0.5 < \tan\beta < 30$ are still allowed by the data. It is interesting to compare to the $M_a$ scan, Fig. 1d, where though the exclusion in $\tan\beta$ is similar, and at low $\tan\beta$ is due to the same final states, at $M_{\text{DM}} > M_a/2 = 125$ GeV the sensitivity at $\tan\beta > 1$ is due to boson+jet rather than dilepton+$E_T^{\text{miss}}$ or $E_T^{\text{miss}}$+jet final states because the decay $a \to \chi\chi$ becomes kinematically closed.

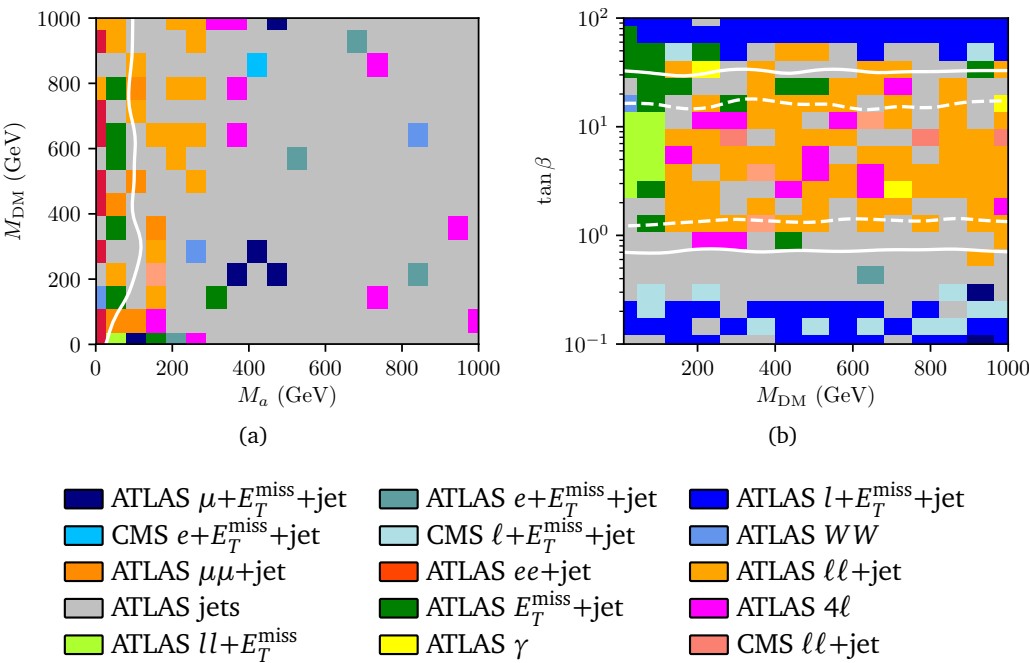

Figure 5: Sensitivity scans in (a) $M_{\text{DM}}$ vs $M_a$ and (b) $\tan\beta$ vs $M_{\text{DM}}$; the colours indicate the most sensitive set of measurements. The 95% (68%) excluded contours are indicated by solid (dashed) white lines; note that the whole plane in $M_{\text{DM}}$ vs $M_a$ is excluded at 68% confidence level. All other parameters are as given in Table 1.

# 6 Varying the mixing parameters

Varying the mixing angle, $\sin\theta$, between the 2HDM pseudoscalar $A$ and the mediator $a$ can change the interplay between different signatures, in particular those that involve top quarks. The DMWG benchmark parameters focus mostly on an intermediate mixing ($\sin\theta = 0.35$).

In this section we investigate instead the case of maximal mixing ($\sin\theta = 1/\sqrt{2}$), as well as scanning down to low values of mixing, where the DM candidate gradually decouples.

First we repeat the scans discussed in Section 3, keeping all parameter choices fixed, with the exception of the mixing angle. The ($M_a, M_A$) scan is shown in Figures 6a,c. The overall sensitivity of SM measurements for this slice of the parameter space is increased towards higher $M_A$ masses for small and intermediate $M_a$ masses. The $ZZ \rightarrow 2\ell + E_T^{\text{miss}}$ measurement [22] still plays an important role for $M_H = M_A < 500$ GeV, especially due to the fact that the $H \rightarrow aZ$ branching ratio increases with $\sin\theta$. For example, at $M_H = 435$ GeV, $M_a = 100$ GeV it approximately doubles as $\sin\theta$ changes $0.35 \rightarrow 1/\sqrt{2}$. In addition the production cross section for the $H$ boson increases from 3.4 pb ($\sin\theta = 0.35$) to 3.6 pb ($\sin\theta = 1/\sqrt{2}$).

A second important analysis that dominates the sensitivity for large $M_A - M_a$ values is the $h \rightarrow \gamma\gamma$ analysis [32]. This analysis selects events where a Higgs boson is radiated from the pseudoscalar mediator $a$, a process whose cross-section is proportional to the mixing and the $M_A - M_a$ mass difference.

The last but not least important signature that acquires additional exclusion sensitivity in the maximal mixing case is the $E_T^{\text{miss}}$+jet final state. Particularly strong sensitivity is obtained by the search for supersymmetric quarks in the 0-lepton channel [11] around $M_A \sim 1$ TeV. This sensitivity arises from the increased gluon-initiated production cross section for $a$ (up to a factor 3-4 for $M_a = 100$ GeV, assuming $M_A = 1$ TeV), as well as the increase in branching ratio of many decays that contribute to the $E_T^{\text{miss}}$+jet final state. In this category we have not only the classic $a$ or $A$ produced in association with a jet and decaying into a DM pair but also $H \rightarrow aZ, H^{\pm} \rightarrow aW^{\pm}$ or $A \rightarrow ah(bb)$.

The ($M_a, \tan\beta$) scan is shown in Figure 6b,d. Also in this case, the $E_T^{\text{miss}}$+jet final states play an important role for small $a$ masses for all $\tan\beta$ values but the overall exclusion sensitivity in this region is a combination of multiple final states with similar sensitivities.

In Figure 7, we show a series of scans from minimal to maximal mixing, for various values of $M_A$ and $\tan\beta$, with $M_a = 400$ GeV.

For intermediate $\tan\beta$, the overall sensitivity shows little dependence on $\sin\theta$, presumably because for these parameters it is not driven by the $E_T^{\text{miss}}$ signature of DM. The $E_T^{\text{miss}}$+jet final state, which is more sensitive at higher BSM Higgs masses, does decline in prominence as the mixing, and hence the DM production cross section, declines. For high and low $\tan\beta$, the sensitivity extends to larger values of $M_A$, and increases with $\sin\theta$, again driven by $E_T^{\text{miss}}$+jet and jet substructure measurements.

# 7 Away from the alignment limit

Finally, the SM Higgs measurements and electroweak fits do still allow some mixing between the SM Higgs and the $H$. Figure 8 shows a scan where $\sin(\beta - \alpha)$ is allowed to move from the alignment limit; for some limited parameter settings (around $0.5 < \tan\beta < 10$ and $M_{H^{\pm}} \geq 600$ GeV) values as low as $\frac{1}{\sqrt{2}}$ are still allowed by the studies in [2]. The combined measurement sensitivity again shows little dependence on $\sin(\beta - \alpha)$, although as we move away from the alignment limit, the diphoton [32, 36] and four-lepton measurements [33, 37] play an increasing role, since these decay channels open up for the $H$. The impact of the measurements of the SM Higgs branching fractions, and of any implied decrease in the SM Higgs cross sections, is not considered here.

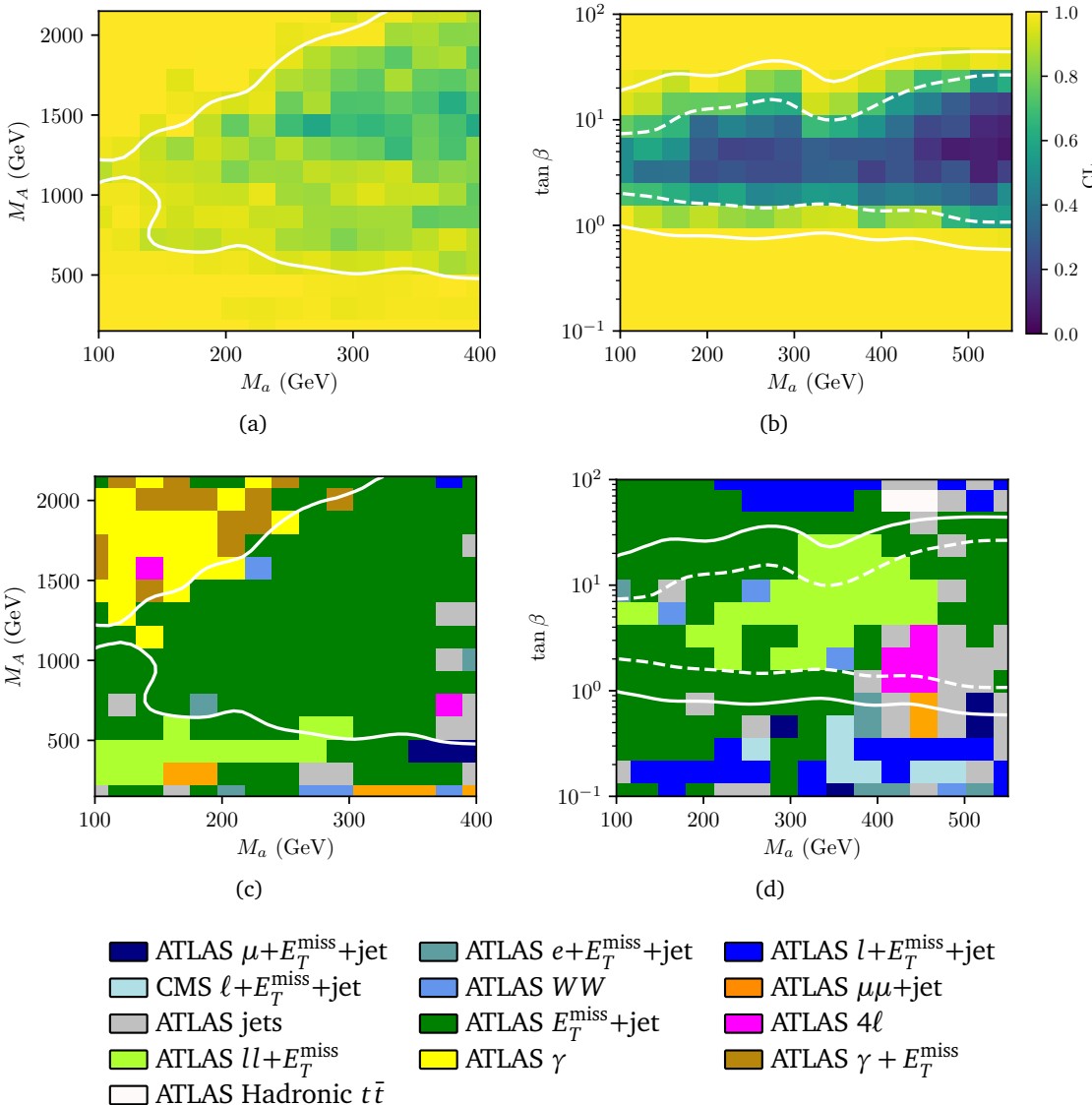

ATLAS $\mu + E_T^{\mathrm{miss}}$+jet · ATLAS $e + E_T^{\mathrm{miss}}$+jet · ATLAS $l + E_T^{\mathrm{miss}}$+jet
CMS $\ell + E_T^{\mathrm{miss}}$+jet · ATLAS $WW$ · ATLAS $\mu\mu$+jet
ATLAS jets · ATLAS $E_T^{\mathrm{miss}}$+jet · ATLAS $4\ell$
ATLAS $ll + E_T^{\mathrm{miss}}$ · ATLAS $\gamma$ · ATLAS $\gamma + E_T^{\mathrm{miss}}$
ATLAS Hadronic $t\bar{t}$

Figure 6: CONTUR scans over the parameter planes from [4] but with the maximal mixing assumption $\sin\theta = 1/\sqrt{2}$ in the (left) $(M_a, M_A)$ and (right) $(M_a, \tan\beta)$ planes. All other parameters are as given in Table 1. The top row shows the sensitivity map in terms of CLs, with 95% (68%) excluded contours indicated by solid (dashed) white lines; note that the whole plane in $M_a$ vs $M_A$ is excluded at 68% confidence level. The bottom row shows the analysis which contributes the most to the sensitivity of each signal point.

# 8 Conclusions

There is interesting sensitivity to the two Higgs doublet plus pseudoscalar DM model across several final states already measured by ATLAS and CMS at the LHC. This is due to the quite complex phenomenology of the model, which can change a lot when the parameters change. The CONTUR approach allows us to efficiently scan a wider range of parameters than usually considered, relaxing some of the current assumptions imposed in benchmark scenarios.

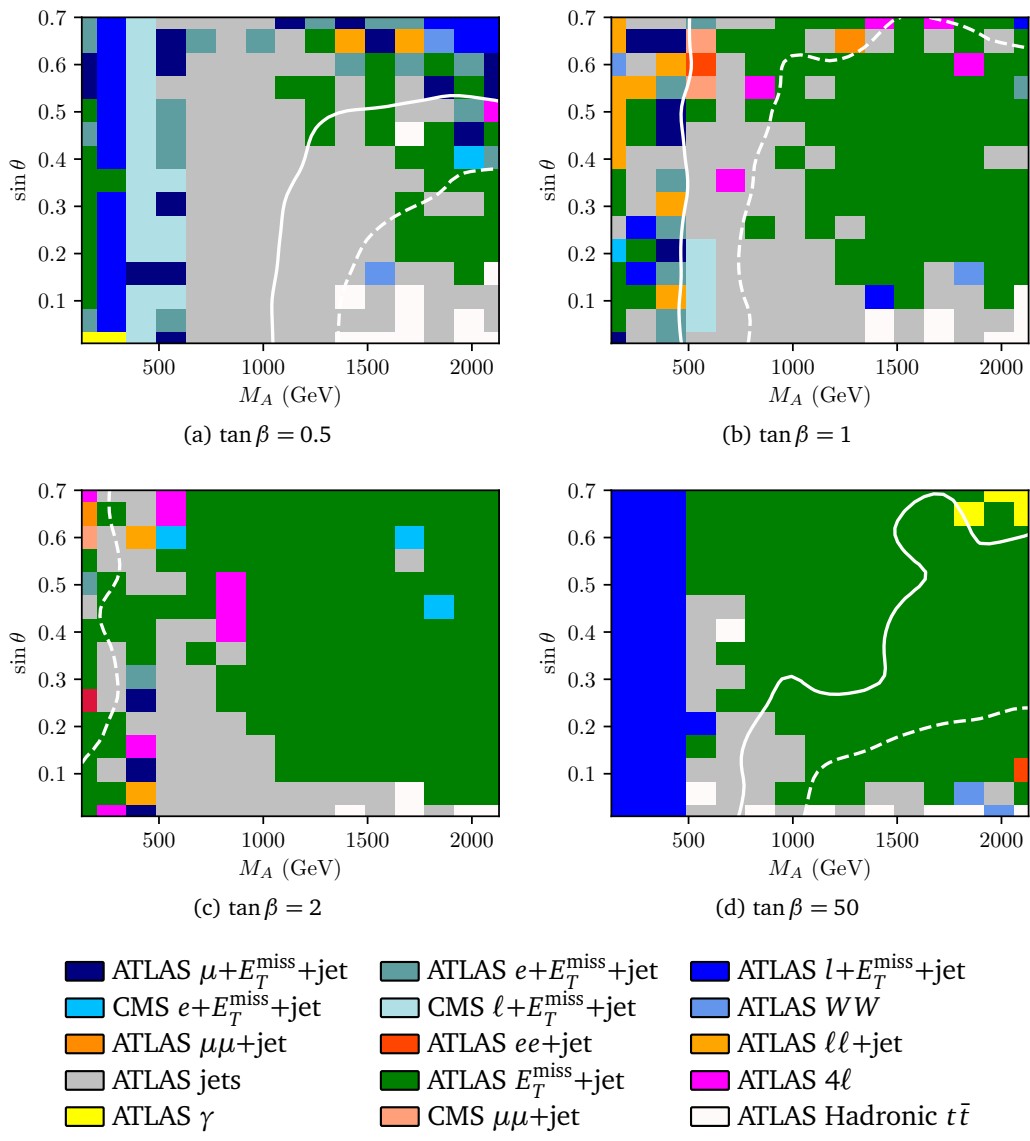

Figure 7: Scans in the common mass of the exotic Higgs bosons and mixing angle $\sin\theta$ between $a$ and $A$ for different $\tan\beta$ with $M_a = 400\,\text{GeV}$. All other parameters are as given in Table 1. The 95% (68%) excluded contours are indicated by solid (dashed) white lines; note that there is no exclusion at 95% confidence level for $\tan\beta = 2$. The colours indicate the most sensitive set of measurements.

Future searches for this model should also consider final states involving top and/or $W$ production, even in the absence of a large missing energy signature. Producing cross section measurements together with such searches would increase the impact and generality of the results, as well as allowing predictions for the relevant SM processes to be probed and tested. At the time of writing, $W$ measurements often resort to using data-driven $b$-jet control regions or applying a $b$-jet veto only at detector level that is therefore not part of the fiducial phase space. This involves extrapolation either of control regions or into unmeasured regions, and in both cases the impact for different model scenarios is unpredictable. If future $WW$ measurements can be made less model-dependent [38], they may also make a significant contribution.

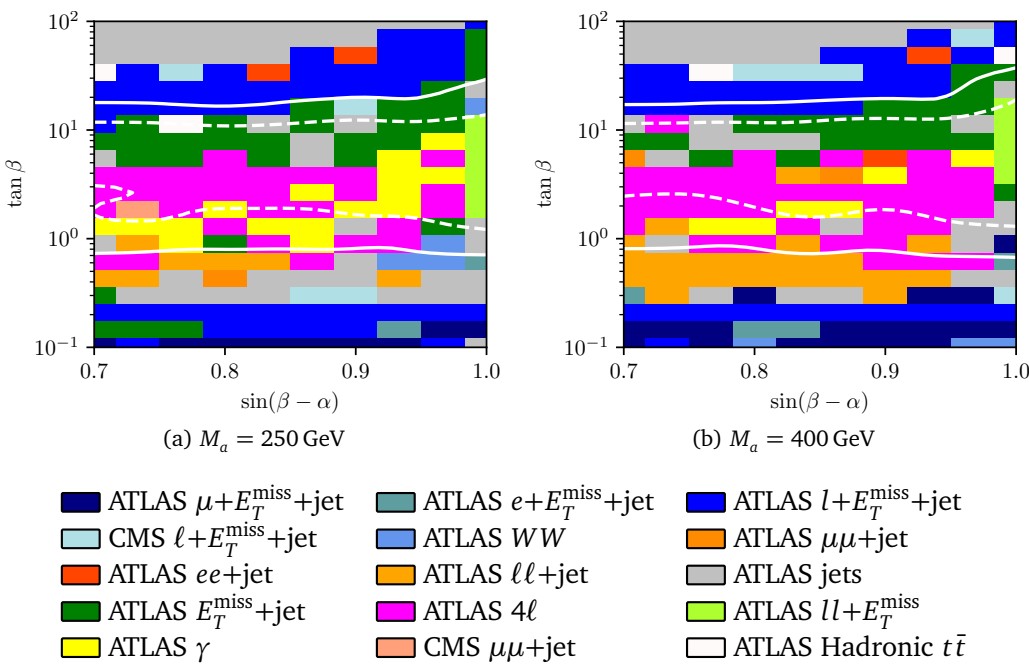

(a) $M_a = 250\,\text{GeV}$  (b) $M_a = 400\,\text{GeV}$

Figure 8: Moving away from the alignment limit of $\sin(\beta - \alpha) = 1$. All other parameters are as given in Table 1. The 95% (68%) excluded contours are indicated by solid (dashed) white lines; the colours indicate the most sensitive set of measurements.

The full run 2, coming run 3, and HL-LHC measurements of a wide variety of final states can be expected to have a substantial impact.

# Acknowledgements

We thank Andy Buckley, Louie Corpe and David Yallup for ongoing CONTUR developments and discussions of the method. We are grateful to the organisers of the 2019 Les Houches workshop on Physics at TeV Colliders, and of the DESY colloquia, where this collaboration began. This work would not have been possible without dedicated computing support from colleagues at DESY and UCL, sustained through the challenges of the COVID-19 lockdown.

**Funding information** JMB has received funding from the European Union's Horizon 2020 research and innovation programme as part of the Marie Skłodowska-Curie Innovative Training Network MCnetITN3 (grant agreement no. 722104) and from a UKRI Science and Technology Facilities Council (STFC) consolidated grant for experimental particle physics. PP's work is supported by the Young Investigator Group research grant from the Helmholtz Association (no. VH-NG-1304). This work used computing equipment funded by the Research Capital Investment Fund (RCIF) provided by UKRI, and partially funded by the UCL Cosmoparticle Initiative.

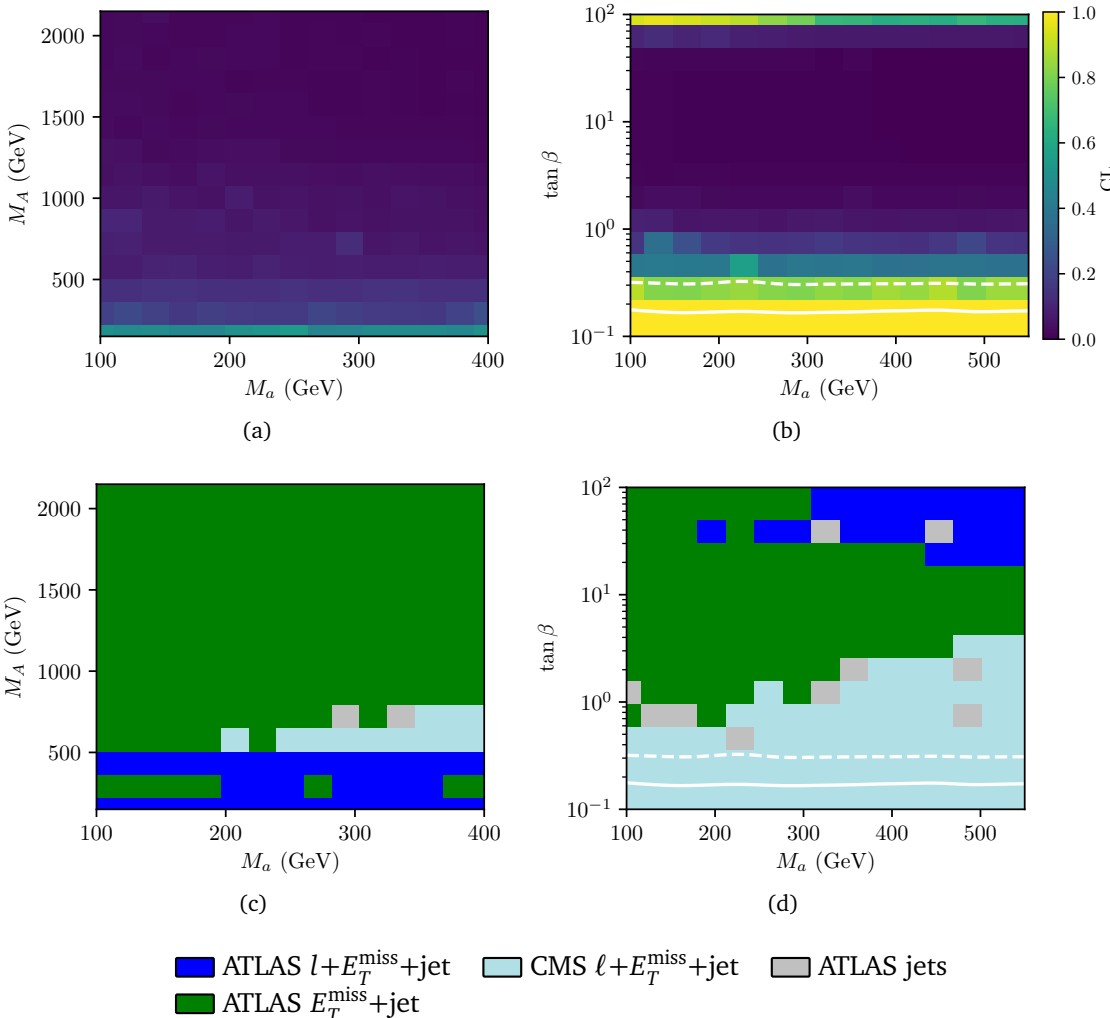

Figure A.1: CONTUR scans over the parameter planes from [4] in the (left) $(M_a, M_A)$ and (right) $(M_a, \tan\beta)$ planes, but considering only $2 \rightarrow 3$ processes generated with Madgraph5_aMC@NLO. The top row shows the sensitivity map in terms of CLs, with 95% (68%) excluded contours indicated by solid (dashed) white lines. The bottom row shows the analysis which contributes the most to the sensitivity of each signal point.

# A    Impact of $2 \rightarrow 3$ processes

As Herwig does not generate $2 \rightarrow 3$ processes, the impact of neglecting those was studied using Madgraph5_aMC@NLO [13]. For this, the CONTUR scans over the parameter planes from [4] shown in Figure 1 were repeated, now only considering potentially significant $2 \rightarrow 3$ processes, namely $pp \rightarrow t\bar{t}X$ and $pp \rightarrow thX$ where $X$ is a BSM particle. The resulting sensitivity is shown in Figure A.1. Main contributions come from $t\bar{t}X$ processes giving rise to the lepton-plus-$E_T^{\mathrm{miss}}$-plus-jet final state that ATLAS and CMS measurements of semi-leptonic $t\bar{t}$ decays are sensitive to, in particular when $\tan\beta$ is small and therefore the coupling of top quarks to the BSM Higgs bosons is large. As $\tan\beta = 1$ is used for the scan in Figure A.1a, the overall sensitivity there is negligible. Minor contributions in both scans come from jets and $E_T^{\mathrm{miss}}$ in the final state. All in

all, the sensitivity to $2 \rightarrow 3$ processes is negligible compared to $2 \rightarrow 2$ and $s$-channel processes and does also not exhibit any shape difference.

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
