# Peer review of "A study of collider signatures for two Higgs doublet models with a Pseudoscalar mediator to Dark Matter"

_SciPost Physics Core, doi:SciPost Phys. Core 4, 003 (2021)_

## Round 2 · Referee Report · Anonymous · 2020-12-17

Report
In this submission the authors study the LHC phenomenology of a model containing two Higgs doublets, a pseudoscalar singlet and a (fermionic) dark matter particle. This model provides a gauge invariant and renormalizable version of pseudoscalar mediation, which is a relevant case among DM simplified models. The authors apply the CONTUR method to scan over two-dimensional planes in the model parameter space and test where the new physics would have significantly contributed to the measurements implemented in the code, which primarily include `SM-like' cross sections, with a few involving $E_T^{\rm miss}$. The study begins from a reference point in parameter space and investigates the effects of varying in turn several model parameters, such as the masses of the BSM Higgs bosons, the DM mass, the mixing between $a$ and $A$, and the mixing between $h$ and $H$.
While CONTUR is an interesting method to test new physics at the LHC, the present manuscript contains a routine (if technically sound) application that I do not expect will have a major impact on future studies of the 2HD + a + DM model. The main reason is that the present analysis does not consider several measurements which are expected or already known to have important impact, examples being mono-Higgs and additional $E_T^{\rm miss}$ measurements concerning Section 4, and measurements of the 125 GeV Higgs properties concerning Section 7. The authors do clearly state these omissions, but the resulting physical picture is lacking key information. In addition, the paper provides limited insight beyond the CONTUR figures, as it presents a series of departures from the initial benchmark but refrains from analyzing any of them in depth. For these reasons, the manuscript clearly does not meet the demanding criteria for acceptance in SciPost Physics.
Nevertheless, in my opinion the paper could be considered for publication in the second-tier journal SciPost Physics Core. It can be viewed as a `CONTUR status report' - an application of this developing framework to an important problem, technically sound and containing some new findings despite the issues mentioned above. If the authors are interested in this possibility, I ask them to address the comments or questions concerning the text listed below.
Requested changes
1) In section 4, line 11, the authors mention that for $M_A = M_{H^\pm}$ there are extra constraints on $\sin\theta$ from electroweak precision, favoring smaller values. Is it obvious that the choice $\sin\theta = 0.35$ made in Section 4.3 is consistent with these constraints?
2) In Section 4.2, fifth-to-last line: when writing `... in this region the $H$ decays dominantly to $ah$', do the authors mean $A\to ah$ perhaps?
3) In Section 4.3, sixth line: `$M_{H^\pm} = M_H$' should read `$M_{H^\pm} = M_A$'. In addition, I was confused by the subsequent sentence: why is $A\to ah$ enhanced at low $M_H\,$? Perhaps the authors mean $A\to aH$, or I am misinterpreting this sentence.
4) In the Conclusions appears the following sentence, `If future $WW$ measurements can be made less model-dependent, for example by including all relevant kinematic cuts ...', which I believe refers to a discussion presented by three of the authors in the Les Houches 2019 report [9], pages 85-88. A reader of the present manuscript is likely to get confused, unless they have already read the Les Houches report. Similarly for footnote 6. These discussions should be slightly expanded to explain these aspects, so the present paper can be read without necessarily consulting [9].
5) A couple of suggestions concerning the Introduction: I think it would be useful to mention explicitly an important reason for interest in pseudoscalar mediation (the fact that single $a$ exchange between DM and quarks only gives rise to suppressed spin-dependent cross section, so direct detection bounds are easily satisfied), and to cite 1404.3716 as an important early study of the 2HD + a + DM model.
6) A few typos: page 3, exotics Higgs bosons $\to$ exotic Higgs bosons; page 6, CP even $\to$ CP-even; on page 11, `... those that couple to top quarks' could be replaced by '... those that involve top quarks' for clarity.

---

## Round 3 · Referee Report · Anonymous (Referee 1) · 2021-2-4

Report

In this resubmitted version the authors have addressed, minimally but satisfactorily, all comments and questions contained in my first report. Therefore, in my view the manuscript is now suitable for acceptance in SciPost Physics Core. There remain just a few typos in the text, listed below, which could be corrected at the proofreading stage.

Requested changes

  • footnote 2 is missing a full stop at its end;
  • both in footnote 6 and in the now-extended Conclusions, in a few cases b-jet $\to b$-jet, WW $\to WW$ and so on;
  • in Ref. [21] the names of some authors are repeated.

---

## Round 3 · Author Response

We thank the referee for the detailed report and helpful comments and suggestions. We have addressed these as discussed below.

We disagree with the referee to some extent, in that our work goes beyond a demonstration or status report of the Contur method (which we agree has already been done). We present new constraints on previously unexplored parameter regions of a high profile model, which we think are of significant interest. However, we accept that this is not a comprehensive survey of all constraints on the model, and if that is what would be required for SciPost Physics, we are content to be published in Scipost Physics Core.

We would like to address the other comments and requested changes in the following: 1) It is not obvious, no, and we should have been clearer, thanks for raising the question. We kept sin(theta)=0.35 as this is a conventional choice by the experiments, but in fact for this value the EW constraints exclude much of the plane for MA above about MA=200GeV with an allowed region at lower MH extending up to MA~1TeV. Our limits exclude the remaining allowed space. We have added a comment and the contours in the corresponding plane. 3) It is indeed meant A→ah here as g_Aah increases with decreasing MH. This can be seen from eq. 4.12 in 1701.07427. 4) We have added some clarification text and a further reference, which we hope addresses this without repeating a large amount of text from other sources. All other points raised, in particular in 2), 3), 5) and 6), are very good spots or suggestions and have been implemented.

---

## Round 3 · List of Changes

Resulting from the discussion above, we have introduced the following changes:

In the introduction, a comment has been added to point out the particular relevancy for pseudoscalar mediation in DM models probed at colliders.

We have added the contours from EW constraints in Section 4, case 3 (MA=MH±) for non-degenerate masses to Fig. 4 and a comment in the corresponding text.

To clarify the problematic model-dependencies of existing WW measurements, we have added a remark in the conclusion and extended the footnote in Section 3.

In addition, we have added further references where appropriate and fixed the typos pointed out by the referee.

---

## Editorial Decision

published